# Modulation of Gut Microbiota and Short-Chain Fatty Acid Production by Simulated Gastrointestinal Digests from Microalga *Chlorella vulgaris*

**DOI:** 10.3390/ijms26062754

**Published:** 2025-03-19

**Authors:** Celia Bañares, Samuel Paterson, Dulcenombre Gómez-Garre, Adriana Ortega-Hernández, Silvia Sánchez-González, Carolina Cueva, Miguel Á. de la Fuente, Blanca Hernández-Ledesma, Pilar Gómez-Cortés

**Affiliations:** 1Institute of Food Science Research (CIAL, CSIC-UAM), Nicolás Cabrera 9, 28049 Madrid, Spain; celia.banares.e@csic.es (C.B.); samuel.paterson@csic.es (S.P.); carolina.cueva@csic.es (C.C.); mafl@if.csic.es (M.Á.d.l.F.); 2Microbiota and Vascular Biology Laboratory, Hospital Clínico San Carlos-Instituto de Investigación Sanitaria San Carlos (IdISSC), C/ Prof. Martín Lagos, 28040 Madrid, Spain; mgomezgarre@salud.madrid.org (D.G.-G.); a.ortega.hernandez@hotmail.com (A.O.-H.); silsangon@outlook.es (S.S.-G.); 3Biomedical Research Networking Center in Cardiovascular Diseases (CIBERCV), Adv. Monforte de Lemos, 3-5, 28029 Madrid, Spain; 4Faculty of Medicine, Universidad Complutense de Madrid (UCM), Plaza Ramón y Cajal, 28040 Madrid, Spain

**Keywords:** *Chlorella vulgaris*, in vitro digestion, colonic fermentation, short-chain fatty acids, gut microbiota

## Abstract

*Chlorella vulgaris* is a source of potential bioactive compounds that can reach the large intestine and interact with colonic microbiota. However, the effects of consumption of this microalga on gastrointestinal function have scarcely been studied. This paper simulates, for the first time, the passage of *C. vulgaris* through the gastrointestinal tract, combining the INFOGEST method and in vitro colonic fermentation to evaluate potential effects on the human colonic microbiota composition by 16S rRNA gene sequencing and its metabolic functionality. The results show that the presence of this microalga increased the release of short-chain fatty acids (SCFAs), such as acetic, propionic, butyric, and isobutyric fatty acids, after 48 h colonic fermentation, being indicators of gut health. In correlation with the release of SCFAs, a significant reduction in bacterial groups causing intestinal imbalance, such as *Enterobacteriaceae*, *Enterococcus* spp., and *Staphylococcus* spp., was observed. In addition, digests from *C. vulgaris* favored intestinal health-related taxa, such as *Akkermansia* and *Lactobacillus*. *C. vulgaris* is, therefore, a promising food ingredient for good intestinal health and the maintenance of a balanced colonic microbiota.

## 1. Introduction

Microalgae have attracted increasing attention in recent years as potential food sources due to their high nutritional value and wide range of applications. Microalgae are rich in proteins, polyunsaturated fatty acids, polysaccharides, pigments, vitamins, and polyphenols [1,2]. Furthermore, the study and use of these matrices represent a great opportunity in terms of revalorization and environmental sustainability due to the ability of microalgae to grow rapidly under different conditions. Thus, microalgae biomass is an interesting alternative to animal products, avoiding the problems associated with their consumption and becoming perfect raw materials in the context of the circular economy [3,4].

Among the large number of available microalgae species, *Chlorella vulgaris* is one of the most studied. This microalga belongs to the generally recognized as safe (GRAS) category and has been endorsed by the European Food Safety Authority (EFSA) as safe for human consumption [5]. For this reason, it is currently marketed in nutraceutical or powder forms [6]. Numerous health benefits of this specie have been reported, such as antioxidant, anti-inflammatory, anticancer, antidiabetic, and immunomodulating effects, among others [4,7], which have been attributed to its rich composition, particularly proteins and peptides, omega-3 fatty acids, chlorophylls, β-carotene, and phycocyanin [8,9]. However, there is still a lack of information regarding the digestibility and bioaccessibility of their compounds along with their impact on the human gut microbiota [10].

In vitro digestion models are versatile and useful tools to evaluate structural modifications, digestibility (circumventing ethical issues related to animal sacrifice), costs, and reproducibility associated with in vivo assays [11,12]. However, achieving physiological conditions in existing in vitro digestion protocols is challenging, especially due to the complexity of setting up these protocols for heterogeneous matrices that vary in macro- and micronutrients, resulting in different digestion products depending on the protocol used [13,14,15]. Within the existing in vitro simulated digestion protocols, despite its limitations, the internationally harmonized in vitro static simulated gastrointestinal digestion protocol INFOGEST [16] has stood out worldwide in the study of digestibility of many different food sources, making it a consensual, reproducible, and efficient starting point for scientists to choose in the study of food digestion. Digestion models applied to microalgae have focused on proteins, lipids, or carbohydrates. Thus, Niccolai et al. (2019) reported that the protein digestibility of *C. vulgaris* was similar to that of proteins from beans and wheat [17]. Canelli et al. (2020) reported that *C. vulgaris*-lipid bioaccessibility could be increased by high-pressure homogenization [2], and Chen et al. (2018) demonstrated that the microalga polysaccharides are not digested in the small intestine and can, therefore, enter directly into the colon and be utilized by the gut microbiota [18]. In addition, Qazi et al. (2021) [19] evaluated the in vitro protein digestibility of raw and ethanol-treated *Tetraselmis chuii*, *Microchloropsis gaditana*, and *C. vulgaris* biomass, either alone or incorporated into wheat bread, following the standardized INFOGEST static in vitro digestion model. However, to the best of our knowledge, no studies on the digestion of the whole microalgae biomass and its impact on gut microbiota have been reported.

The gut microbiota has been shown to be involved in important functions, such as protection of the intestinal epithelial barrier, regulation of the immune system, and protection against pathogens [10,20]. It is also involved in metabolic and nutritional functions, such as the fermentation of non-digestible dietary waste into short-chain fatty acids (SCFAs). Some of these SCFAs, such as butyrate, acetate, and propionate, are implicated in glucose and lipid metabolism, epithelial cell differentiation, and anticancer properties [21,22].

Regarding the effect of microalgae species on gut microbial ecology, the work of Ashraf et al. (2022) demonstrated the different interaction dynamics between microalgae and their associated microbiomes, suggesting that such interactions could analogously benefit gut health by promoting microbial diversity, supporting digestion, and producing metabolites like SCFAs that, overall, improve host health [23]. Moreover, in the Huang et al. (2023) work, the addition of *C. vulgaris* in juvenile Nile tilapia (*Oreochromis niloticus*) diets promoted the diversity of their gut microbiota by increasing the alpha and beta diversity as well as the abundance of specific beneficial microbial groups like *Reyranella*, supporting the ability of microalgae to interact with gut microbiota [24].

Although several compounds present in microalgae biomass have been reported to potentially modulate the human gut microbiota, the studies are still scarce. Recently, Barros de Medeiros et al. (2021) demonstrated the promoting effect of microalgae-derived oligosaccharides and phenolics of different microalgae species [25]. They showed the ability of *C. vulgaris* biomass to increase the abundance of *Bifidobacterium* spp. and that other microalgae would exert a modulatory effect on the intestinal microbiota, increasing the abundance of beneficial microorganisms (*Lactobacillus–Enterococcus*). On the other hand, it is also important to consider the effect of microalgae in metabolic functionality of the gut microbiota. In this sense, protein-enriched microalgae extracts may promote a greater production of SCFAs during colonic fermentation [25]. Considering the limited information available about the interaction of *C. vulgaris* with gut microbiota, the main objective of the present work was to evaluate the impact of simulated gastrointestinal digests from microalga *C. vulgaris* on the gut microbiota composition (plate counting and 16S rRNA gene sequencing) and metabolic functionality (SCFAs and ammonium production).

## 2. Results and Discussion

### 2.1. Analysis of Microbial Communities

As a first approach, the effects of the *C. vulgaris* and inulin gastrointestinal digests on the colonic microbiota were evaluated by plate counting. From a microbiological point of view, differences in values were considered significant when they were ≥1 log colony forming units (CFU)/mL. Figure 1 shows the results of microbial populations expressed as the log CFU/mL for total aerobes and anaerobes obtained at the different incubation times (0, 24, 48, and 72 h). For total aerobes (Figure 1A), a substantial increase in microbial counts was observed after 24 h in the presence of non-absorbable fraction (NAF) from the *C. vulgaris* digest (4.25 vs. 8.13 log CFU/mL at time 0 h) and of inulin (4.34 vs. 8.07 log CFU/mL at time 0 h). Then, samples in the presence of *C. vulgaris* NAF showed a progressive decrease in aerobes down to 3.54 log CFU/mL at 72 h, while they remained constant until the end of the colonic fermentation of inulin NAF (8.25 log CFU/mL at 72 h). Regarding total anaerobes (Figure 1B), the trend from the beginning to the end of the colonic fermentation was very similar for both microalga and inulin, keeping the microbial counts practically constant at around 8.0 log CFU/mL.

To understand the influence on the most representative bacterial groups of the gut microbiota, Figure 2A–E depict the microbial counts of *Enterobacteriaceae*, *Enterococcus* spp., lactic acid bacteria, *Clostridium* spp., and *Staphylococcus* spp. in the presence of NAF from the *C. vulgaris* and inulin digests. Most of the microbial populations increased after 24 h of colonic fermentation, indicating an adaptation of the bacteria to the culture medium. Then, the decreases observed at 48–72 h incubation time indicate that nutrients were depleted, and therefore, the largest changes in microbial population take place. The samples obtained at the end of the *C. vulgaris* NAF colonic fermentation generally showed lower values of the microbial counts in comparison with the samples obtained after colonic fermentation of inulin NAF. In the case of the microbial counts of *Enterobacteriaceae*, *Enterococcus* spp., and lactic acid bacteria (Figure 2A–C), they were substantially lower in *C. vulgaris* compared to inulin. Previous studies have shown that the growth of *Enterobacteriaceae* members produces an imbalance in the microbiota that leads to a persistent state of inflammation in bowel diseases [26]. Therefore, the decrease in this bacterial group might result in a beneficial effect at the intestinal level.

Regarding *Clostridium* spp. (Figure 2D), the results of *C. vulgaris* were very similar to those for inulin. However, Barros de Medeiros et al. reported a relative decrease in *Clostridium* spp. when non-digested *C. vulgaris* biomass was added to the medium [1]. As for *Staphylococcus* spp., an increase was observed at 24 h (5.11 log CFU/mL) in the presence of the microalgae NAF (Figure 2E), but, in general, no stimulation was observed, and a decrease was seen after 48 h and 72h. This would be positive since *Staphylococcus aureus* has often been related to the appearance of different diseases, and its presence in the intestine can increase the risk of intestinal infection [27]. These results agree with previous research in which *C. vulgaris* showed antimicrobial effects, significantly reducing the growth of *Staphylococcus aureus* [28,29].

### 2.2. Short-Chain Fatty Acid Analysis

SCFAs are metabolites produced by the intestinal microbiota during the colonic fermentation process that can contribute to host physiology and energy homeostasis [30]. The main SCFAs associated with microbial metabolism are acetic, propionic, and butyric acids, produced in a molar ratio of 3:2:1, respectively [31]. Their release has been linked to numerous beneficial effects on human health [32]. A sharp increase in the production of these SCFAs at 24 h of fermentation was observed in *C. vulgaris* samples, and their levels were maintained at 48–72 h (Figure 3). It is important to highlight that the microalga NAF promoted greater production of SCFAs by the gut microbiota than inulin during the fermentation process. Figure 3A shows the evolution of the concentration of acetic acid after the fermentation with the NAFs obtained from the *C. vulgaris* digests, evidencing a significantly favored release in comparison to inulin. Acetic acid increased progressively until the end of the fermentation, mainly at 24 and 48 h. This fatty acid has important functions in the body, as it is a building block for tissues and a substrate for cholesterol synthesis [33]. Regarding the release of propionic acid (Figure 3B), significant differences were detected between the microalga NAFs and inulin at 24 h. This fatty acid is associated with reduced tissue sensitivity to insulin and lowers serum cholesterol levels [10]. Interestingly, microalgae showed the greatest impact on the release of butyric acid (Figure 3C), with more than a 4-fold increase at 24 h, which was maintained until the end of the fermentation (2.94 mM at 72 h). These values were significantly higher than those determined with inulin (0.55 mM and 1.48 mM after 24 h and 72 h, respectively). This fatty acid is an important source of energy for colonic epithelial cells. It may also be involved in alleviating insulin resistance in diabetes, reducing inflammatory processes, and promoting satiety [34,35]. In addition, microalgae fermentation significantly promoted the release of isobutyric acid (Figure 3D) compared to inulin. Different studies have shown that the release of isobutyric acid can modulate glucose and lipid metabolism in the liver, similar to the major SCFAs, contributing to improving insulin sensitivity in individuals with impaired metabolism [36].

### 2.3. Protein Degradation and Ammonium Production

During the simulated gastrointestinal digestion, pepsin and pancreatin only represented 1.67% of the protein content. According to the protein value reported in our previous article for the NAF (34.73%) [37], in 3 g of this fraction used for the colonic fermentation assay, 1.04 g corresponded to protein (1.023 g of microalga protein and 0.017 g to enzymes). Thus, the contribution of enzymes to the protein content of the fraction used in the colonic fermentation assay was not significant. As a measurement of microbial proteolytic activity, Figure 4 shows the protein content and ammonium production during the colonic fermentation of NAF from the *C. vulgaris* digests and inulin. The protein concentration significantly decreased at 24 h of the fermentation process (Figure 4A). From 0 to 72 h of fermentation, on average, it accounted for 52.7% and 38.8% for *C. vulgaris* and inulin, respectively. This reduction in protein concentration over time could be due to the use of microalgae proteins as a substrate of microbiota during microbial fermentation [38]. Figure 4B shows an increase in the ammonium concentration over fermentation time for all the samples studied. Specifically, the ammonium levels were increased by 12.7 and 5.9 times for *C. vulgaris* and inulin NAFs, respectively, at the end of the microbial fermentation. Ammonium production was statistically different between inulin and the microalga specie during the whole fermentation process.

Ammonium release is an indicator of protein metabolism, and its level is considered a clue to the wellness of the microbial population [39]. In general terms, ammonium production depends on the substrate administered as well as the duration of the treatment [40]. The decrease in the protein content and the consequent generation of ammonium during the fermentation of *C. vulgaris* indicated that microalgae protein is an optimal source of nitrogen for the intestinal microbiota [41]. Thus, the increase in ammonium release in the presence of a microalga fermentation substrate would suggest that *C. vulgaris* is an easily metabolizable food ingredient.

### 2.4. 16S Ribosomal RNA Gene Analysis

To evaluate the impact of the simulated gastrointestinal digestion of *C. vulgaris* on the gut microbiota composition and compare it with that exerted by inulin digest, we examined the effect on the bacterial community using 16S rRNA gene sequencing after 48 h of colonic fermentation. First, we calculated α-diversity, a parameter that considers the intrinsic biodiversity of each sample by assessing the richness and evenness of the samples (Figure 5A–E). For this purpose, we rarefied the sequence depth to 197,479 read counts in all samples. As expected, the blank sample at zero time was not different from all samples neither in richness (observed OTUs and Chao1) nor in evenness (Pielou, Shannon, and Simpson indexes). After 48 h of fermentation with both NAFs of *C. vulgaris* and inulin, α-diversity indexes changed significantly, indicating a lower level of bacterial richness and community evenness. Several studies have reported a decrease in bacterial abundance during fermentation [42,43], which has been associated with the competitive effect of dominant strains inhibiting the growth of others. However, the samples incubated with NAF from *C. vulgaris* at 48 h had significantly higher observed OTUs and Chao1 indexes than those with inulin but comparable community evenness. This suggested a differential effect of *C. vulgaris* in the α-diversity with respect to inulin.

Regarding the β-diversity, which considers the biodiversity among groups, no differences from all samples were found at zero time (Figure 5F). After 48 h fermentation, the samples of both C. vulgaris and inulin NAFs differed from the zero time in the bacterial composition but with significant differences in the microbiota composition of *C. vulgaris* samples in comparison to those of inulin. The first two principal coordinates presented over 80% of the bacterial community variation (Figure 5F). These results showed that the fermentation of *C. vulgaris* had a specific effect on the gut microbiota community, reducing the α-diversity and changing the gut microbial composition.

To identify how *C. vulgaris* affected the fecal microbiota, a taxonomic analysis was carried out. In terms of relative abundance, the dominant phyla at zero time were Bacteroidota (≈46%), Firmicutes (≈38%), and Proteobacteria (≈13%) in all samples (Figure 5G). After the fermentation, both *C. vulgaris* and inulin NAFs decreased the abundance of Bacteroidota and increased Proteobacteria phyla. A drastic decrease in Firmicutes phylum was also observed after *C. vulgaris* and inulin fermentation (Figure 5G). Interestingly, the *C. vulgaris* samples but not inulin showed higher levels of Fusobacteriota, Desulfobacterota, and Verrucomicrobiota phyla (Figure 5G,H). The decrease in Bacteroidota and the increase in Proteobacteria phyla aligned with the observed production of acetic acid that lowered the pH [44]. However, the differential effect on the other phyla demonstrated a distinct impact of *C. vulgaris* compared to inulin.

To further characterize the taxonomic differences, we performed Kruskal–Wallis tests at the genus level. We identified 59 genera showing significant differences in abundance between *C. vulgaris* samples and inulin NAFs (FDR-corrected *p* < 0.05) (Figure 6). At the genus level, the *C. vulgaris* samples were classified into one cluster, and the inulin samples were classified into another one. At 48 h, both the *C. vulgaris* and inulin samples were characterized by a higher abundance of Gram-negative bacteria, many of them belonging to Proteobacteria phylum (*Clostridium sensu stricto 1*, *Parabacteroides*, *Vicinamibacteraceae*, *Enterobacter*, *Escherichia/Shigella*, *Citrobacter*, *Klebsiella*, and *Salmonella*). However, the relative abundance of these opportunistically pathogenic bacteria was significantly lower in the *C. vulgaris* samples than in those of inulin. Additionally, the *C. vulgaris* samples showed an increment of the SCFA-producer bacteria *Akkermansia*, *Dialister*, *Lactobacillus*, *Anaerostipes*, *Butyricicoccus*, *Butyricimonas*, *Coprococcus*, *Phascolactobacterium*, and *Ruminococcus* with respect to inulin (Figure 6). Specifically, LEfSe analysis (Figure 7) identified *Akkermansia*, *Dialister*, *Lactobacillus*, *Anaerostipes*, *Butyricicoccus*, *Butyricimonas*, *Coprococcus*, *Phascolactobacterium*, and *Ruminococcus* as differentially abundant genera (LDA score over 2.5) in the *C. vulgaris* samples. By contrast, *Citrobacter* and *Escherichia/Shigella* were differentially abundant genera (LDA score over 2.5) in the inulin samples (Figure 7). The diverse population of bacteria in the *C. vulgaris* and inulin samples could account for specific associations that produce different SCFAs.

Two important health-related bacteria were identified as biomarkers in *C. vulgaris* colonic fermentations. The *C. vulgaris* samples were characterized by a significant increment in *Akkermansia* that could modulate host metabolism by producing SCFAs and influencing energy metabolism, which might contribute to its protective effects against metabolic disorders such as obesity and diabetes [45]. Additionally, by metabolizing mucin, *Akkermansia* might help to maintain the integrity of the gut barrier and regulate immune responses in the gut. The *C. vulgaris* samples were also characterized by a significant increment in *Lactobacillus*. An important mechanism by which *Lactobacillus* can confer health benefits is through their ability to colonize and compete in the gut, thereby helping to maintain a healthy balance of intestinal microbiota [45]. Additionally, *Lactobacillus* is one of the main producers of acetic acid, which can lower intestinal pH and create a less favorable environment for the growth of pathogens [46]. This SCFA can also strengthen the intestinal barrier and help prevent harmful bacteria from adhering to intestinal epithelial cells. Furthermore, certain *Lactobacillus* strains have been suggested to modulate the host’s immune response, thereby enhancing immune function and reducing inflammation in the gut [47]. These mechanisms contribute to the beneficial effects of *Lactobacillus* on gastrointestinal and overall health.

Furthermore, the composition of the absorbable fraction and NAF from the *C. vulgaris* digests could potentially explain the gut modulation properties observed. Thus, as we previously reported [48], the NAF had a higher content of total phenols than its corresponding absorbable fraction (28.43 ± 1.94 vs. 36.91 ± 2.66 mg gallic acid equivalents/g fraction). A higher protein content in NAF (34.73 ± 2.80%) than in the absorbable fraction (23.74 ± 1.75%) was also observed. In addition, within microalgae species, *C. vulgaris* is considered to have a thick and polysaccharide-rich cell wall, mainly made of glucosamine [49], suggesting that some of its components could be present in the NAF. Therefore, the phenolic compounds potentially bound to the undigested polysaccharides and proteins in the microalga NAF may interact with gut microbiota after digestion. This interaction could produce energy that supports primary microbial consumers and their syntrophic partners while also highlighting the well-documented gut-modulating effects of phenolic compounds.

Moreover, our findings agreed with recent preclinical animal works. Thus, the study of Velankanni et al. (2023) demonstrated that supplementation with *C. vulgaris* altered the gut microbiota composition and increased SCFA production in mice [50]. Nishimoto et al. (2021) [51] showed that individuals with low concentrations of fecal propionate showed an increase in propionate concentration upon *C. pyrenoidosa* intake. In addition, recent clinical studies regarding *C. vulgaris* biological effects have focused on type 2 diabetes patients [52]. Since these studies used *C. vulgaris* extracts instead of the whole biomass, information on the effects of microalgal biomass on the human gut microbiota remains scarce.

In comparison with other GRAS microalgae with biotechnological potential such as *Chlamydomonas reinhardtii*, Fields et al. (2020) [53] examined, for the first time, the effects of consuming the whole biomass of this specie in both mice and humans, with an emphasis on gut health. They found that the addition of *C. reinhardtii* biomass to the diet of healthy humans had a beneficial impact on gastrointestinal function, although no significant changes in the microbial composition were observed. Although the translation of in vitro outcomes to in vivo effects remains an open question, our findings highlighted *C. vulgaris* as a promising GRAS microalga due to its gut health benefits. Overcoming one of the limitations of this study—the lack of a negative control using water instead of microalgal digestate or inulin during colonic fermentation—would allow for the determination of the presence of endogenous bacteria and, consequently, a more reliable interpretation of the results, enabling confirmation of the beneficial role of the microalga in gut health.

Despite providing valuable insights into food digestion and microbial metabolism, in vitro digestion and fermentation models have significant limitations in fully replicating the complexity of the human gastrointestinal tract. One major drawback is their inability to mimic the dynamic conditions of the gut, including peristalsis, enzyme variability, and fluctuating pH levels. For instance, a review by Nadia et al. (2024) [54] highlighted the challenge of establishing in vitro–in vivo correlations, emphasizing that in vitro models often fail to accurately replicate gastric and intestinal digestion kinetics. Moreover, although the use of intestinal organoids is emerging as an alternative in vitro model for studying gastrointestinal digestion, Medvedeva et al. (2024) [55] discussed the limitations of organoid and explant models, noting that crypt-derived organoids fail to maintain epithelial integrity and do not fully replicate the inflammatory environment of the gut. Additionally, in vitro fermentation models often rely on simplified microbial communities that do not capture the full diversity and metabolic interactions present in the human colon [56]. Therefore, further studies are needed to study *C. vulgaris* biomass behavior during gastrointestinal digestion to potentially identify responsible compounds for the observed effects and to elucidate their mechanisms of action.

## 3. Materials and Methods

### 3.1. Samples and Reagents

Commercial microalga *C. vulgaris* biomass was kindly supplied by AlgaEnergy S.A. (Madrid, Spain). The nutritional composition of the biomass in terms of total proteins (61.27 ± 0.46 g/100 g biomass), lipids (10.20 ± 2.00 g/100 g biomass), and carbohydrates (6.64 ± 0.65 g/100 g biomass) has been previously described [48]. Pepsin (EC 232-629-3; 3200 units/mg protein), pancreatin (232-468-9; 8X USP), bile salt, potassium persulfate (K_2_S_2_O_8_), monosodium phosphate (NaH_2_PO_4_), disodium phosphate (Na_2_HPO_4_), monopotassium phosphate (KH_2_PO_4_), calcium chloride (CaCl_2_), hydrochloric acid (HCl), sodium hydroxide (NaOH), sodium chloride (NaCl), yeast extract, K_2_HPO_4_, NaHCO_3_, MgSO_4_·7H_2_O, CaCl_2_·6H_2_O, Tween-80, hemin, vitamin K, L-cysteine, and phosphoric acid (H_3_PO_4_) were purchased from Sigma-Aldrich (St. Louis, MO, USA). These reagents were used to prepare the corresponding simulated gastric fluid (SGF), simulated intestinal fluid (SIF), and colonic fermentation medium for the consequent in vitro gastrointestinal digestion and colonic fermentation following Brodkorb et al. (2019) [16] and Tamargo et al. (2023) [57] indications.

### 3.2. Simulated Gastrointestinal Digestion

The simulated gastrointestinal digestion was based on the INFOGEST static protocol, with slight modifications [16]. For the simulation of the oral phase, 1 g of dried microalga biomass was dissolved in 500 µL of Mili-Q water, and 4.5 mL of human salivary fluid was added [58]. The mixture was then incubated at 37 °C at 130 rpm for 2 min in the Environmental Shaker—Incubator ES 20/60 (Biosan Medical-biological Research & Technologies, Warren, MI, USA). After completing the oral phase, 4.8 mL of SGF was added to the mixture, adjusting the pH to 3.0 with 1 M HCl. Next, 3 µL of 0.3 M CaCl_2_ solution and 300 µL of pepsin (ratio enzyme:substrate, E:S, 1:100, *w*/*w*) were added. The simulation of the gastric phase was performed by incubation of the mixture at 37 °C for 2 h. The orogastric digest was mixed with 5.1 mL of SIF, and the pH was adjusted to 7.0 with 1 M NaOH. Then, 1.5 mL of bile (1:50, *w*/*w*), 24 µL of 0.3 M CaCl_2_ solution, and 3 mL of pancreatin (E:S 1:3, *w*/*w*) were added. After 2 h of incubation at 37 °C, the enzymes were inactivated by heating (95 °C, 5 min) in a Memmert thermostatized bath (Schwabach, Germany), and the digest was centrifuged at 2000× *g* for 30 min at 4 °C in an EppendorfTM Centrifuge 5804R (Hamburg, Germany). Two fractions were obtained: the supernatant corresponding to the absorbable fraction that was previously used to analyze the phenolic compounds, peptide profile, and antioxidant activity [48]; and the precipitate corresponding to the NAF that was collected. Four digestion replicates were conducted, and the NAFs obtained were pooled and stored at −20 °C until the colonic fermentation assays were carried out. *C. vulgaris* underwent two simulated gastrointestinal digestions on two different days that were analyzed separately, although the data were averaged to be presented in the text and in the figures. One g of inulin, a compound with a known prebiotic effect [59,60], was digested following the same simulated protocol, although the complete inulin digest was used during the colonic fermentation phase to assure the presence of the prebiotic compound. This sample, digested in quadruplicate, was used as a control.

### 3.3. Static Colonic Fermentation

The fecal inoculum was prepared by combining 1 g of feces obtained from a healthy volunteer [61] with 10 mL of phosphate-buffered saline (PBS) solution at a concentration of 0.1 M and pH 7.0, and homogenized using a Stomacher 400 Circulator (Seward Inc., Bohemia, NY, USA) to prepare a fecal slurry [62]. The fermentation flasks were prepared according to Tamargo et al. (2023) with slight modifications [57]. Briefly, 6 mL of the fecal inoculum were mixed with 54 mL of colon nutrient medium (CNM, peptone water (2 g/L), yeast extract (2 g/L), NaCl (0.1 g/L), KH_2_PO_4_ (0.04 g/L), NaHCO_3_ (2 g/L), MgSO_4_·7H_2_O (0.01 g/L), CaCl_2_·6H_2_O (0.01 g/L), Tween 80 (2 mL/L), hemin (0.05 g/L), vitamin K (10 μL/L), L-cysteine (0.5 g/L), bile salts (0.5 g/L), and distilled water) and 3 g of the NAF obtained after microalga digestion. In parallel, incubations with a known prebiotic compound (inulin) were carried out. All fermentation flasks were incubated for 72 h and 120 rpm simulating the conditions of the distal region of the human large intestine (pH 6.8, 37 °C and anaerobic atmosphere) [62]. Fermentation assays were carried out in triplicate and samples were collected at 0, 24, 48, and 72 h. An immediately collected sample (1 mL) was used for microbial counts. Others two aliquots of 2 mL were centrifuged at 10,000 rpm for 10 min at 4 °C; the supernatants were filtered through a 0.22 µm filter and stored at −20 °C for SCFA and ammonium analysis, while the pellets were stored at −80 °C for 16S ribosomal RNA gene analysis.

#### 3.3.1. Microbial Plate Counting

Immediately after each sampling step, tenfold serial dilutions of each colonic samples were plated on different types of media as described in Jiménez-Arroyo et al. (2023) [63]. Plate counting was done in triplicate, and data were expressed as log CFU/mL. Differences in values were considered significant when they were higher or lower than 1 log (CFU/mL) compared to the inulin (control).

#### 3.3.2. Short-Chain Fatty Acid (SCFA) Analysis

The fermentative activity of colonic microbiota was assayed by determining the production of SCFAs. Before analysis, 50 µL of defrosted sample were acidified with 0.5% phosphoric acid (200 µL) and mixed with 100 µL of internal standard (2-methylvaleric acid, Sigma-Aldrich, 1.97 mM). The mixture was extracted with 1 mL of n-butanol and the analysis was carried out by gas chromatography coupled to flame ionization detector, according to the methodology described by García-Villalba et al. (2012) [64]. A gas chromatograph (Agilent, Santa Clara, CA, USA) equipped with a DB-WAX capillary column (100% polyethylene glycol, 30 m, 0.325 mm internal diameter × 0.25 μm thickness, Agilent) was used. Helium was used as carrier gas at a flow rate of 1.5 mL/min. The oven temperature program was as follows: the initial temperature was 50 °C for 2 min, then a first ramp of 15 °C/min until reaching 150 °C, followed by a second ramp of 5 °C/min up to 200 °C and a final ramp at 15 °C/min to 240 °C, which was maintained for 20 min. The total analysis time was 41.3 min. The injector and detector temperatures were 250 °C and 260 °C, respectively. For SCFA quantification, a mixture of standards (Ref. FLPK-005K, Agilent) was used to prepare calibration curves of each SCFA in the concentration range from 0.005 to 30 mM. The internal standard 2-methylvaleric acid was also spiked to the standard mix to correct any bias related to sample preparation or chromatographic analysis. Quantitative data were obtained by calculating the area of each compound relative to the internal standard. These analyses were carried out in triplicate.

#### 3.3.3. Protein and Ammonium Content

The proteolytic activity of colonic microbiota was assayed by measuring the protein content and the production of ammonium ion (NH_4_^+^) in the supernatants obtained after colonic fermentation. The protein concentration of each sample was carried out by the bicinchoninic acid (BCA) method using the Pierce BCA kit (Thermo Fisher Scientific, Waltham, MA, USA) following the protocol previously described by Paterson et al. (2023) [65]. Bovine serum albumin (BSA) at concentrations ranging from 50 to 1000 µg/mL was used as a standard. The NH_4_^+^ production was measured using the Photometric Spectroquant^®^ ammonium reagent test (Merck & Co., Kenilworth, NJ, USA), measuring the absorbance at 690 nm using the Biotek SynergyTM HT plate reader (Winooski, VT, USA). The results were obtained by interpolation into the calibration curve constructed using standard solutions between 2 and 75 mg NH_4_^+^/L. Supernatants were diluted with deionized water to adjust their concentration to the kits’ measurement range, and the results were expressed as mg NH_4_^+^/L. Both ammonium and protein analyses were carried out in triplicate.

### 3.4. 16S Ribosomal RNA Gene Analysis

#### 3.4.1. Extraction and Quantification of Microbial DNA

The microbial DNA from the pellets obtained at fermentation times of 0 and 48 h for *C. vulgaris* and inulin was extracted using the QIAamp Fast DNA Stool mini kit (Quiagen, Hilden, Germany). DNA integrity was evaluated with the BioAnalyzer 2100 (Agilent), and the concentration was quantified with the fluorimeter Qubit 3.0 using the dsDNA HS assay (Life Technologies S.A., Alcobendas, Madrid, Spain).

#### 3.4.2. 16S Ribosomal RNA Gene Sequencing and Bioinformatics Analysis

Each colonic sample underwent amplification of the 16S ribosomal RNA gene via polymerase chain reaction (PCR) using the Ion 16S Metagenomics kit (Life Technologies S.A.), designed to amplify seven hypervariable regions (V2, V3, V4, V6–7, V8, and V9). Subsequently, libraries were generated from 5 ng of DNA per sample by repairing the ends of the amplicons with the Ion Plus Fragment Library kit (Life Technologies S.A.), and attaching DNA molecular identifiers using the Ion Express Barcode Kit Adapters (Life Technologies S.A.). Following library preparation, libraries were diluted to 22 pM and underwent clonal amplification by emulsion PCR in the Ion OneTouchTM 2 System (Thermo Fisher Scientific). Finally, all samples were sequenced using an Ion S5TM System with an Ion 520TM Chip (Life Technologies S.A.). Bioinformatic analysis was performed using the QIIME 2™ platform [66]. The obtained sequences were clustered into operational taxonomic units (OTUs) with a 99% of coincidence with the SILVA v.138 database. α-Diversity, obtained from a rarefied OTUs profile, was assessed with five metrics: observed OTUs, Chao1 richness estimate, Shannon diversity index, Pielou’s evenness index and Simpson’s diversity index. For β-diversity analysis, we calculated Jaccard dissimilarity index, with distances between groups and samples plotted using principal coordinate analysis (PCoA). A linear discriminant analysis (LDA) effect size (LEfSe) was used to measure the relative abundances of taxa in the analyzed groups [67].

### 3.5. Statistical Analysis

Data of SCFAs, protein and ammonium content are expressed as mean ± standard error mean (SEM), and α-diversity indexes as median (interquartile range). Differences among groups were analyzed with a one-way ANOVA followed by a Tukey test, a Kruskal–Wallis test or a paired *t*-test as appropriate. The differences between groups regarding the β-diversity (structure of the bacterial communities) were analyzed with a permutation analysis and multiple ANOVA (PERMANOVA) with 999 permutations. LEfSe consists of the application of a Kruskal-Wallis test to identify taxa with significantly different relative abundances, followed by LDA score to determine an effect size of each taxon. The threshold of LDA score was less than 2.5. The heat map consists of the application of R package ANCOM-BC2 v.2.4.0 [68] to obtain significantly different abundances between algae and inulin. Benjamini-Hochberg false discovery rate correction (FDR) was used to adjust *p*-value for multiple testing, and FDR < 0.05 was used as a significance threshold. Statistical analysis was performed with STATA IC16.1 (Stata-Corp LLC, TX, USA), GraphPad Prism 8.0 (GraphPad Software, San Diego, CA, USA), the scipy PYTHON package and IBM^®^ SPSS^®^ Statistics software package version 27 (IBM Inc., Armonk, NY, USA). For all analyses, differences were considered statistically significant when *p* < 0.05.

## 4. Conclusions

The present study demonstrated that the digests of *C. vulgaris* increased the levels of acetic, propionic, butyric and isobutyric acids after 48 h of in vitro fermentation. Changes in SCFAs production could be associated to changes in the microbiota composition. It is particularly noteworthy that *C. vulgaris* induced an increase in the abundance of bacteria associated with host health and simultaneously inhibited the relative abundance of opportunistically pathogenic bacteria, which were more significant than those of inulin. Our data demonstrated that *C. vulgaris* could become a practical food ingredient to enhance health, and to hinder disease, by improving the intestinal microbial environment.

## Figures and Tables

**Figure 1 ijms-26-02754-f001:**
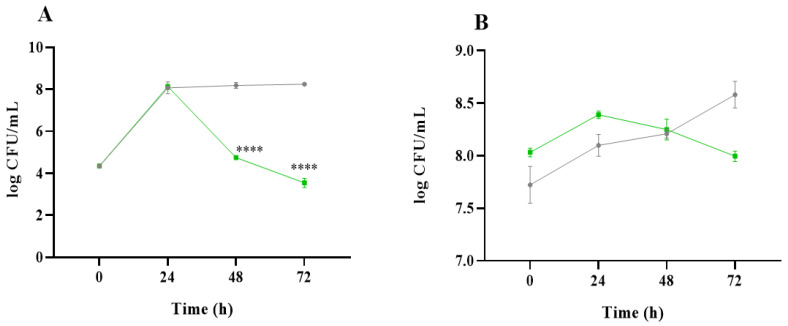
Evolution of total aerobe (**A**) and anaerobe (**B**) microorganisms (log CFU/mL) at different times of colonic fermentation in the presence of non-absorbable fraction (NAF) of *Chlorella vulgaris* and inulin during 0, 24, 48, and 72 h. *C. vulgaris* (green) and inulin (control, grey). For the statistical analysis, differences in values were considered significant (****) when they were higher or lower than 1 log (CFU/mL) compared to the inulin (control).

**Figure 2 ijms-26-02754-f002:**
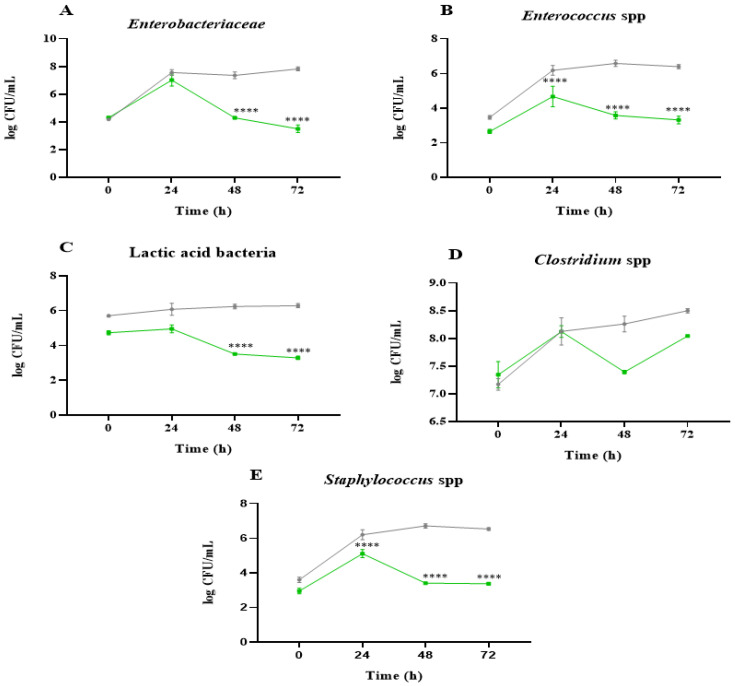
Effect of the non-absorbable fraction (NAF) from *Chlorella vulgaris* and inulin during 0, 24, 48, and 72 h on growth of microorganisms of gut microbiota (log CFU/mL). (**A**) *Enterobacteriaceae*, (**B**) *Enterococcus* spp., (**C**) Lactic acid bacteria, (**D**) *Clostridium* spp., (**E**) *Staphylococcus* spp. *C. vulgaris* (green) and inulin (control, grey). For the statistical analysis, differences in values were considered significant (****) when they were higher or lower than 1 log (CFU/mL) compared to the inulin control.

**Figure 3 ijms-26-02754-f003:**
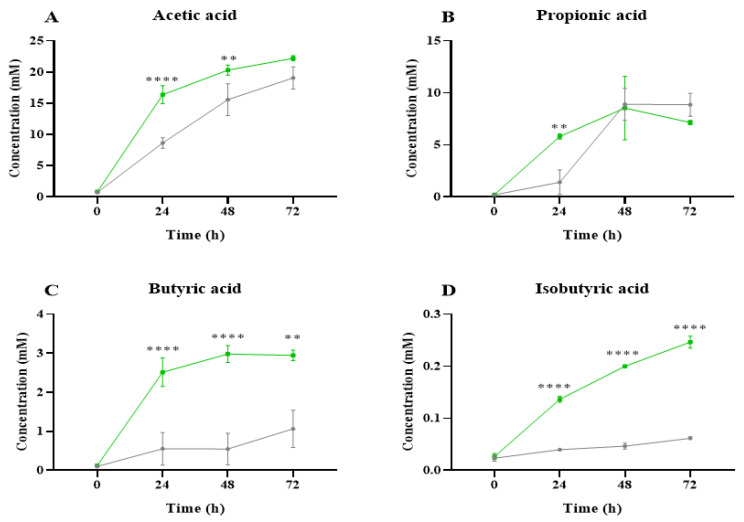
Short-chain fatty acid (SCFAs) production during colonic fermentation (0, 24, 48 and 72 h) of the non-absorbable fraction (NAF) of *Chlorella vulgaris* and inulin. (**A**) Acetic acid, (**B**) Propionic acid, (**C**) Butyric acid, (**D**) Isobutyric acid. *C. vulgaris* (green) and inulin (control, grey). The statistical analysis was performed with GraphPad Prism 8.0 (GraphPad Software, San Diego, CA, USA) using a one-way analysis of variance (ANOVA) followed by a Tukey test ** *p* ˂ 0.01; **** *p* ˂ 0.0001 level of significance at a particular time compared to the inulin control.

**Figure 4 ijms-26-02754-f004:**
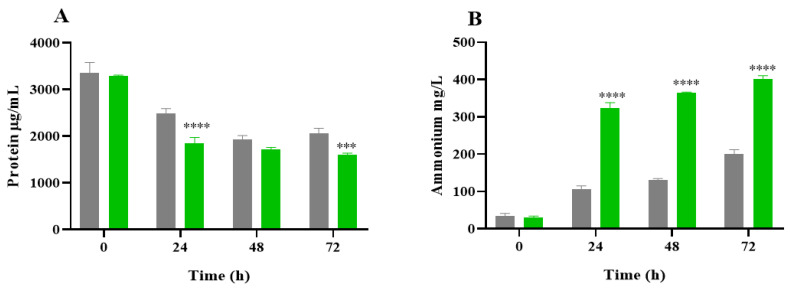
Protein content (**A**) and ammonium production (**B**) during colonic fermentation (0, 24, 48, and 72 h) of the non-absorbable fraction (NAF) of *Chlorella vulgaris* and inulin. *C. vulgaris* (green) and inulin (control, grey). The statistical analysis was performed with GraphPad Prism 8.0 (GraphPad Software, San Diego, CA, USA) using a one-way analysis of variance (ANOVA) followed by a Tukey test. *** *p* ˂ 0.001; **** *p* ˂ 0.0001 level of significance at a particular time compared to the control.

**Figure 5 ijms-26-02754-f005:**
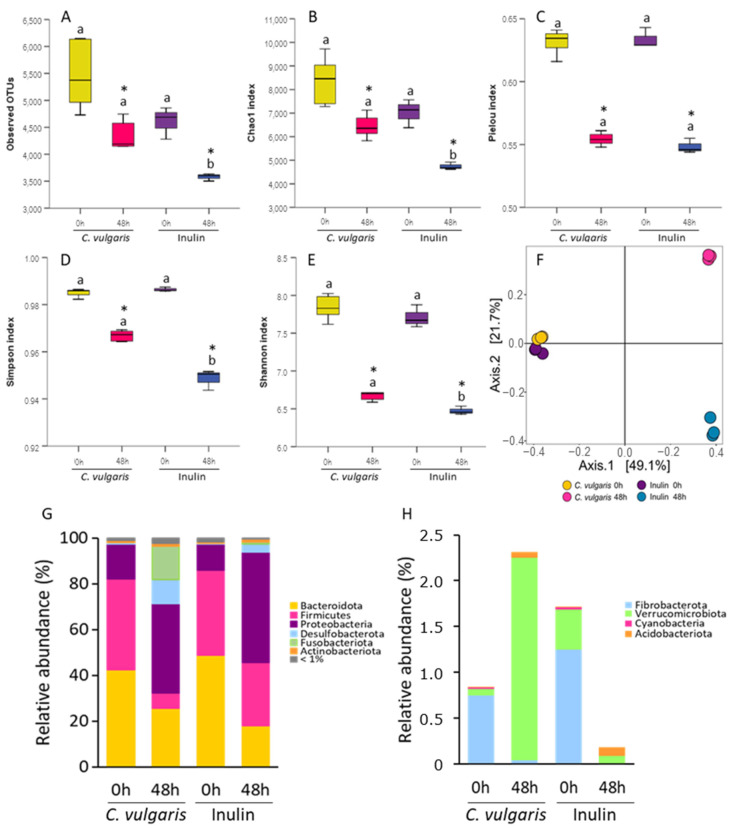
Effect of the non-absorbable fraction (NAF) from *Chlorella vulgaris* and inulin on α- and β-diversity, and taxonomic composition of gut microbiota at genera level. α-diversity is represented by observed operational taxonomic units (OTUs), (**A**), Chao1 index (**B**), Pielou’s evenness index (**C**), Shannon index (**D**), and Simpson index (**E**). The results are shown in box plots. Different letters indicate significant differences at each time point. * indicate significant differences compared to time zero. β-diversity (**F**) is represented by a principal coordinate analysis (PCoA) plot of Jaccard index dissimilarity (PERMANOVA *p* < 0.05). Abundance of majority (**G**) and minority bacterial genera (**H**).

**Figure 6 ijms-26-02754-f006:**
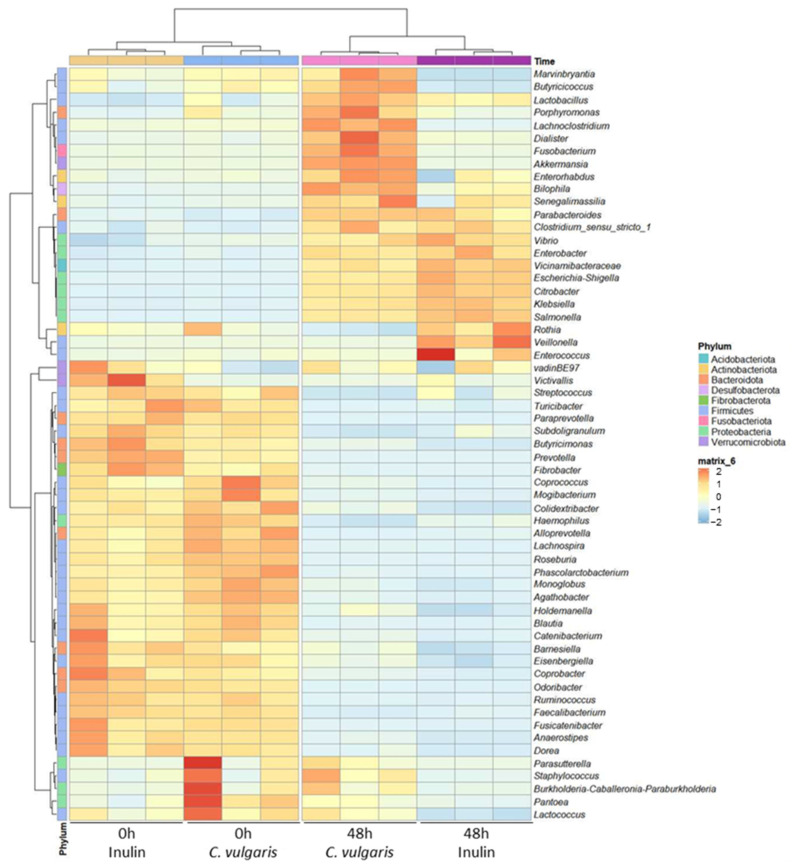
Heatmap diagram of the gut microbiota composition at genus level of the non-absorbable fraction (NAF) from *Chlorella vulgaris* and inulin. The 59 genera that showed significant differences in abundance between *C. vulgaris* samples and inulin are displayed from less abundant (blue) to more abundant (red). The samples are represented by columns, and the microbial taxa are represented by rows. It has been used hierarchical clustering based on Euclidean distance using the average method.

**Figure 7 ijms-26-02754-f007:**
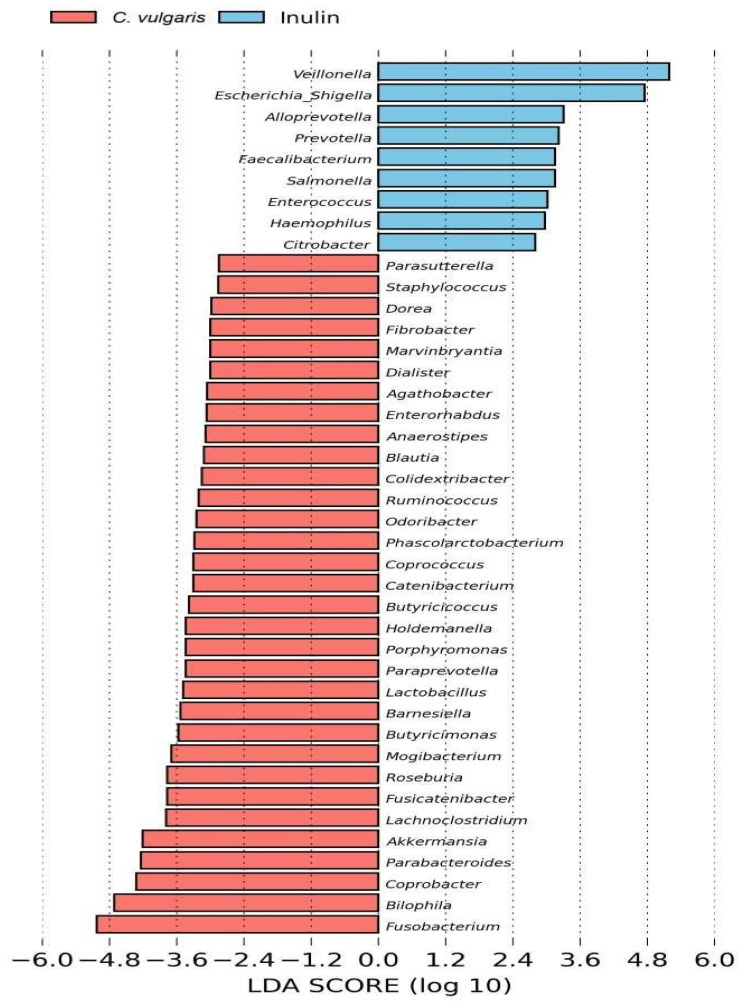
Effect size analysis (LEfSe) showing genera differentially expressed. The length of the horizontal bars (*C. vulgaris* in red bars and inulin in blue bars) represents the linear discriminant analysis (LDA) score (threshold log LDA score ≥ 2.5).

## Data Availability

The original contributions presented in this study are included in the article. Additionally, raw data and metadata have been deposited with the BioProject database at the National Center for Biotechnology Information (NCBI), Available online: https://www.ncbi.nlm.nih.gov/sra/?term=PRJNA1218495 (accessed on 1 February 2025). Further inquiries can be directed to the corresponding authors.

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
