# Peer review of "Modulation of Gut Microbiota and Short-Chain Fatty Acid Production by Simulated Gastrointestinal Digests from Microalga Chlorella vulgaris"

_ijms, 2025, doi:10.3390/ijms26062754_

Round 1

Reviewer 1 Report

Comments and Suggestions for Authors

Celia Bañares et al reported the modulation of gut microbiota composition and function by simulated gastrointestinal digests from Microalga Chlorella vulgaris in vitro. The topics of great interest and the data are clearly presented. My comments are as follows:

  1. The authors are suggested to discuss effects and reported data of vulgaris in human and in preclinical animal models, as well as in other in vitro models.  
  2. The authors are suggested to discuss correction of the in vitro observation from this study with the in vivo function of vulgaris, and the observation from other in vitro models.
  3. How can ensure the function of simulated gastrointestinal digest if the complete inulin digest was used during the colonic fermentation phase? Do the authors have a negative control, incubation with vehicle control, in this study?
  4. A brief description of the statistical methods should be included in the Figure legends.

Reviewer 2 Report

Comments and Suggestions for Authors

Dear Authors,

-I believe this is a good study with important data. However, I think it has a critical limitation: the absence of a negative control in the results. Specifically, why wasn't an incubation performed with only water (no Chlorella or inulin) to determine the presence of endogenous bacteria? I believe this control is essential for confidently interpreting the results obtained with Chlorella

Majors:

-L57: “In vitro digestion models” I find the introduction well-written, but I miss references to studies on other microalgae and a more detailed explanation of the types of in vitro digestion models and the reason for choosing the one used.

-L117: “a mixture of saliva from 7 healthy volunteers” That sounds odd, are there any references to support this? How was the human saliva fluid prepared and standardized?

-L120: “simulated gastric fluid (SGF)” What is the composition and origin?

-L124: “simulated intestinal fluid (SIF)” What is the composition and origin?

-L144: “1 g of feces obtained from a healthy volunteer” why only one? Why not several and then mixed? Please explain

-In the section on short-chain fatty acid (SCFA) analysis, provide more details about the internal standard used. Include its concentration, source, and the reason for choosing it.

-Key point. I don't understand why a negative control was not used in the results. That is, why one of the incubation was not performed without Chlorella or inulin to determine the presence of endogenous bacteria. I believe this control is critical to confidently interpret the results obtained with Chlorella.

-I suggest expanding the discussion on the specific mechanisms by which microalgae compounds (e.g., polysaccharides, polyphenols) can modulate the gut microbiota.

-The discussion analyzes the findings in the context of existing literature but could be strengthened by directly comparing the results with previous studies on the effects of other microalgae species on the gut microbiota. “Chlorella vulgaris is one of the most studied species” Yes is correct, but there are also others, such as Chlamydomonas, which is also GRAS. And several recent reviews on its biotechnological potential have been published. Could this result be extrapolated to Chlamydomonas or other algae? Please discuss.

-Discuss in greater depth the limitations of using in vitro digestion and fermentation models to fully replicate the complexity of the human gastrointestinal tract

Minors:

L28:  in-dicators. Tipographic

-L132: “activity24” Tipographic

Round 2

Reviewer 1 Report

Comments and Suggestions for Authors

The authors have addressed the majority of the questions, and the manuscript is now ready for publication.

Author Response

Thanks for the last revision

Reviewer 2 Report

Comments and Suggestions for Authors

Dear Authors,

I believe the authors have adequately addressed all of my comments and suggestions, and I accept the paper in its current version.

Author Response

Thanks for the last revision of the manuscript